# Use of Different Cooling Methods in Pig Facilities to Alleviate the Effects of Heat Stress—A Review

**DOI:** 10.3390/ani10091459

**Published:** 2020-08-20

**Authors:** Dorota Godyń, Piotr Herbut, Sabina Angrecka, Frederico Márcio Corrêa Vieira

**Affiliations:** 1Department of Cattle Breeding, National Research Institute of Animal Production, Balice n Kraków, 31-047 Kraków, Poland; 2Department of Rural Building, Faculty of Environmental Engineering and Land Surveying, University of Agriculture in Krakow, 31-120 Kraków, Poland; p.herbut@ur.krakow.pl (P.H.); s.angrecka@ur.krakow.pl (S.A.); 3Biometeorology Study Group (GEBIOMET), Universida de Tecnológica Federal do Paraná (UTFPR), Estrada para Boa Esperança, km 04, Comunidade São Cristóvão, Dois Vizinhos PR 85660-000, Brazil; fredericovieira@utfpr.edu.br

**Keywords:** cooling, heat stress, pigs, welfare, physiological indicators

## Abstract

**Simple Summary:**

Heat stress triggers behavioural and physiological response and thus negatively affects productivity in livestock. Among farm species, pigs are especially susceptible to exposure to high air temperatures. The evaporative pad cooling system, high pressure fogging, sprinkling as well as elevating of air velocity and floor cooling are the methods most often used in pig facilities to alleviate the effects of heat stress. This article mainly discusses issues linked with the effectiveness of the various cooling methods on improving the indices of animal welfare.

**Abstract:**

An increase in the frequency of hot periods, which has been observed over the past decades, determines the novel approach to livestock facilities improvement. The effects of heat stress are revealed in disorders in physiological processes, impaired immunity, changes in behaviour and decreases in animal production, thus implementation of cooling technologies is a key factor for alleviating these negative consequences. In pig facilities, various cooling methods have been implemented. Air temperature may be decreased by using adiabatic cooling technology such as a high-pressure fogging system or evaporative pads. In modern-type buildings large-surface evaporative pads may support a tunnel ventilation system. Currently a lot of attention has also been paid to developing energy- and water-saving cooling methods, using for example an earth-air or earth-to-water heat exchanger. The pigs’ skin surface may be cooled by using sprinkling nozzles, high-velocity air stream or conductive cooling pads. The effectiveness of these technologies is discussed in this article, taking into consideration the indicators of animal welfare such as respiratory rate, skin surface and body core temperature, performance parameters and behavioural changes.

## 1. Introduction

Bearing in mind the important issues of animal welfare it must be stated that the optimal microclimatic conditions for livestock is a main factor which provides for the animal’s comfort and health. Thus, the proper environmental conditions may also lead to maintain a suitable level of animal production. The main elements creating the microclimate in a facility for farm animals are: air temperature, relative humidity and air speed, as well as the concentration of harmful gases and other pollutants. The lighting and solar exposure of the building are also significant [1,2,3]. Taking into consideration the intensive pig farming system, the mechanical ventilation system seems to be of key importance to providing a thermal neutral zone for these animals especially during summer [4]. In countries of hotter climate, lightweight buildings with sidewall curtains are preferred for pigs’ maintenance [5,6]. This type of building ensures natural cross–ventilation. However, the increase of ambient temperature, which has been a global concern, calls for new methods and additional technology to improve cooling of the animals during summer. The heat stress conditions is a major factor which affect animal health and production, especially in modern genetic lines [7,8].

There are various environmental modifications which may enhance heat exchange in pigs. The systems may cause a decrease of air temperature in a building or cause direct cooling of an animal’s skin. Some of these applications use the effect of water evaporation (misting, fogging, sprinkling, evaporative pads, drip cooling). Heat loss in pigs during hot weather may also be improved by implementation of methods enhancing convection or conduction heat exchange. In pig housing, technologies based on increasing air velocity such as zone cooling and tunnel ventilation have often been used [5,9]. Especially for individually kept sows at the phase of pregnancy or lactation, the enhancing of conductive heat exchange by using floor-cooling technologies may also contribute to alleviating the consequences of heat stress [8,10].

Changes in parameters of animal physiology, behaviour and productivity have been widely considered as indicators of animal welfare [11,12,13,14]. They have often been taken into consideration together with monitoring of microclimate parameters—mainly air temperature and humidity.

The objective of this paper is to present the actual state of knowledge of the most widely used methods of cooling pigs and to show the effectiveness of these technologies on improvement of animal welfare indicators. According to the knowledge of the authors of this article, the more comprehensive approach to the impact on the animal’s body of the functioning of various cooling technologies is not very common and not often found in the world literature on this species.

## 2. Thermoregulation, Behavioural and Physiological Changes in Pigs Affected by High Ambient Temperature

### 2.1. Thermoregulation

In homoeothermic animals, heat production resulting from metabolic processes must be exchanged within a body (across cellular and vascular membranes), and between a body and its environment [15]. The ability of regulation of core body temperature, regardless of environmental conditions, is achieved by both behavioural and autonomic mechanisms. It is widely known that the preoptic area of the hypothalamus plays an important role in coordinating autonomic thermo effectors [16]. In turn, mechanisms and pathways of nervous systems linked with activating the behavioural thermoregulation are still not so clear. It is known that skin, as it represents the interface between the body and the environment, plays a significant role in evoking behavioural thermoregulatory mechanisms [16,17]. Taking into consideration exposure of the body to high ambient temperature, it must be stated that the importance of autonomic mechanisms in thermoregulation is irrefutable; however, this depends directly on body resources, such as water. Due to this fact, the expression of thermoregulatory behaviour is essential to strengthen the body’s response to high air temperatures [16].

In thermoneutral environmental conditions the processes of heat production and its removal from the body are kept in balance, however it is always a dynamic equilibrium. It involves continuous and variable metabolic heat production (cellular metabolism and cell work) and convective and conductive heat transport through tissues [15]. Eventually the heat is removed from the body mainly through radiation, conduction, convection and evaporation, but in the thermoneutral zone the last mentioned way is kept to a minimum [18]. Evaporation of water (through the skin or respiratory tract) and evaporation of sweat are essential factors in removing heat when the organism is challenged with higher ambient temperatures. Skin’s cutaneous vasodilatation is activated in this situation, leading to a decrease of core-to-shell and shell-to-ambient thermal gradients. The consequence is the heat-gain reduction of the body core, but further heat dissipation must be enhanced by an increase of evaporation processes [15,16].

### 2.2. Behavioural and Physiological Changes

The provision of proper housing conditions may cause a possibility of demonstration of whole forms of behavioural thermoregulation and thus provide enhancing of animal comfort. Under high air temperatures pigs present a decrease in their activity, lying in cold and humid areas and increased wallowing behaviour. When kept outside they may also exhibit shade-seeking. Moreover, the heat-stressed animals increase water intake and reduce feed intake. Limited contact with the pen mates is also observed [12,13,19,20,21,22]. An increase of respiratory rate, body-core and skin-surface temperature as well as a reduction of voluntary feed intake are markers which are often taken into consideration in studies of heat-stressed pigs [12,13,23]. Under normal environmental conditions, the respiratory rate ranges from 20 to 40 breaths/min [11,12,14]. The normal body temperature in resting piglets is about 39.5 °C, in finishing pigs 39.3 °C, in gilts 38.8 °C and in multiparous sows it is about 38.3 °C [24]. The average skin-surface temperature is about 33.5 °C in pigs kept in a thermoneutral zone [25].

When discussing the effect of high ambient temperatures, it should be stated that there are two types of challenging situations. Exposition to ambient heat for a short period and for a longer time. It has been shown that there is a biphasic profile of response to the heat load [26]. Within the first 24–48 h after exposure of high ambient temperature (short-term heat acclimation), a rapid increase of respiratory rate and rectal temperature occurred. The second phase of acclimatisation was characterised by a gradual decrease and subsequently reached relative constant levels of these parameters [26]. The acute-stress response was evaluated by Sapkota et al. [25]. The authors showed that after 30 min of exposing pigs to heat load (air temperature of 39.3 °C) the body-core temperature (measured in digestive tract) increased from 38.3 °C (neutral conditions) to 40.3 °C. The authors also found that average skin-surface temperature increased from 33.5 °C to 40.5 °C, but in contrast to the body-core temperature, it dropped when the animals were moved back to the thermal neutral conditions. In turn, de Oliveira et al. [27] studied both effect of acute (48 h) and chronic stress (71 d) on finishing pigs. Short-term heat stress caused an increase of 0.6 °C in rectal temperature; in turn pigs’ exposure to heat for a longer period was characterised by a body core temperature 0.2 °C higher than normal values. The respiratory rate in this study was also higher in acute (125.2 breaths/minute) than in chronic (86.4 breaths/minute) stress.

Due to a lack of functional sweat glands, increased evaporation through the respiratory track is the only effective way to dissipate heat load in pigs [22,28]. Huynh et al. [12] showed rapid increase of respiratory rate at the air temperature above 22.4 °C in finishing pigs. This was convergent with the findings of other studies which evaluated the thermal comfort of lactating sows [29,30]. The optimum temperature for this group was established between 16 °C and 22 °C. In the study of Huynh et al. [12], air temperature was raised from 16 °C to 32 °C. The authors showed phases of acclimatisation; firstly the respiration rate was found to increase as well as the water-to-feed ratio, then decreased heat production and feed intake was observed followed by an increase of rectal temperature.

The reduction of voluntary feed intake in heat-stressed animals is the one of the most important adaptation processes [22,23,30]. The thermic effect of feeding is one of components of total heat production [31]. Additionally, the other portions of this sum, such as basal metabolism and activity heat production are generally at a high level in modern pigs [32]. Thus, the decrease of food intake and the increase of inactive forms of behaviour seems to be a natural way of endogenous heat reduction. The scale of decline depends on the breed, body weight, physiological status, sex and environmental factors [23]. In an earlier study it was shown that high ambient temperature compared with thermoneutral environmental conditions (30 °C vs. 20 °C) may cause a drop of feed consumption in growing and finishing pigs at the level of 50% [33]. Collin et al. [34], taking into control young pigs kept for almost 2 weeks in climatic chambers, found a 30% decrease in feed intake at 33 °C compared to individuals kept in thermoneutral conditions. A similar percentage of decrease may be calculated from the other research when comparing voluntary feed intake in finishing pigs kept at temperatures from 16 °C to 26 °C and at a temperature of 32 °C [12]. In turn, maintaining growing pigs at air temperatures of 35 °C for 7 d resulted in decrease of 47% in feed intake compared to individuals kept in a thermoneutral zone (20 °C) [35].

### 2.3. Molecular and Cellular Response to Heat Stress in Pigs

It is widely known that a reduction of food intake has negative implications on animal production [28,34]. However, the latest studies have shown evidence that heat stress directly induces processes at proteomic and genomic levels [36]. Nowadays the molecular biology methods including high throughput genomic approaches shed new light on processes associated with the effect of heat on cell structure and function [36,37,38,39]. Briefly, it may be characterised that heat causes changes of gene expression, leads to oxidative damage and changes the intracellular signal transduction [40,41]. The recent study of Ma et al. [42] using transcriptone analysis of the longissimus dorsi showed that heat load causes downregulation of the gene involved in muscle structure and development, energy and catabolic metabolism. In turn, it upregulates genes mainly involved in DNA or protein damage/recombination, or processes of cell cycle, biogenesis and stress and immune responses. The other study showed that heat stress induces hepatic protein expression related with heat-shock protein response, oxidative stress response and immune defence [36]. Moreover, based on genome-wide association studies (GWAS) it has been identified that some genomic regions are related to feeding activity in heat-stressed pigs [39]. The studies show that the base of changes in feeding behaviour are primarily genes involved in immune response and function. The same approach (GWAS) was also lately used in the study of Kim et al. [43] to evaluate the genomic base of the physiological indicators of heat stress such as respiration rate, rectal temperature and skin temperature.

When discussing molecular/cellular response to the heat-load exposure, it may be generally concluded that it is divided into different phases. Primarily, the response is based on heat-shock protein expression, next interferon-inducible genes and then activation of small nonspecific stress responses of specific cell lines [44].

Heat-shock proteins (HSPs) are considered as one of the most important heat-stress markers [45]. The increased expression of HSPs have been observed in such tissues as the gut, liver, muscle or ovary of heat-stressed pigs [36,46,47]. This marker may be also easily determined by serum through commercially available tests [48,49]. The main role of HSP is inhibition of apoptosis. They act in protein folding, trafficking and protein complex assembly or disassembly. It has also been shown that polymorphisms in some regions in the HSP70 gene are linked with heat tolerance in livestock [50]. It is also worth mentioning that heat-shock proteins have important implication in modulation of the immune system and are a significant factor in metabolism regulation [51,52]. Moreover they are taken into consideration in studies of extracellular vesicles—vehicles of intercellular signalling [41].

The development of biomarkers linked with molecular and cellular response and the generally expanding knowledge on topics related to biological processes induced by heat stress with use of advanced genomic approaches may certainly be used as a novel means of striving for better strategies to alleviate the negative environmental impact in livestock species. However, in studies presented in this review, respiratory rate, measurements of temperature, assessment of morphological or biochemical markers as well as behavioural changes were the basic indicators of evaluating the effectiveness of cooling technologies.

## 3. Cooling Technologies

Various environmental modifications may be implemented to improve convection, conduction, radiation and evaporative heat loss in pigs during hot ambient temperatures. Moreover, to enhance the effect of animal comfort during hot weather, a combination of different cooling methods is often used. It concerns especially technologies combining the cooling effect of forced-air velocity together with water evaporation.

### 3.1. Cooling Methods Based on Water Evaporation

Fogging, misting or showers as well as evaporative pads have been used in pig housing [6,11,53]. These methods may significantly improve heat loss of nonsweating animals [54]. Water droplets sprayed (depending on their size) either fall on a small pen area (showers) or cause wetting of a larger surface (misting). In both these cases the animal skin is wetted. Drop cooling is another method to directly cool the animal surface. The nozzles of this system are often applied to individual pens above the animal’s neck. Drop cooling causes the release of water droplets with fairly large intervals over time (e.g., 2 L/h) [9]. All the aforementioned methods causes vapourisation of water from the animal body surface, and thus improve evaporative heat loss in pigs.

In turn using high-pressure fogging systems, >5 MPa, or a fan and pad system (evaporating pads) cause the effect of adiabatic air cooling. In brief, this is a process in which energy exchange occurs as a result of contact of air molecules with water molecules. The sensible heat transferred from air to water covers only the demand for energy necessary for the evaporation process [55]. The migration of thermal energy associated with the transition of water into a gas significantly contributes to the reduction of air temperature (dry bulb temperature) [56]. These methods lead to a higher temperature gradient and thus improved sensible heat dissipation [54]. Evaporative pads are designed to cool the air before it will get into the building. They may have large surfaces (from the side or gable walls of the building) or smaller dimensions [6,57]. The smaller surface evaporative pads are often used to cool air conducted (negative pressure) through plastic ducts with holes on it or with inlets above the animal’s head (snout cooling) [6,10]. Larger pads are often used with tunnel ventilation [58]. The general rule of evaporative pads is that of a fan operation forcing the flow of warm air through water-soaked material; it is usually impregnated cellulose paper, shaped in a way that allows molecules of water to mix with air [59,60]. As the higher air speed is essential to increase the effectivity of the pad cooling system, its role in mitigating the effect on heat stress in pigs will be discussed in more detail in the section referring to cooling technologies based on high forced-air velocity.

Taking into consideration all the cooling methods based on water evaporation, it is worth mentioning that the drawback of these technologies may be an increase of air relative humidity. As researchers have shown, increased humidity causes serious difficulties in terms of excess heat from the animal’s body [12,18]. Moreover, there are factors such as saturation deficit, water vapour partial pressure and fluid surface in contact with the air which significantly determine evaporative fraction and time constant for evaporation of a fluid. These factors, however, are also determined by air temperature, relative humidity, quality and temperature of the liquid, droplet radius and air movement [54]. Haeussermann et al. [61] pointed out that the advantages of using a fogging system in comparison to direct cooling (sprinkling of the animal skin) are seen mainly in the lower water amount used in this system as well as the fact that evaporation can still occur even when the indoor air humidity is high. In turn, the other authors [62], when comparing the effectiveness of using fogging and evaporative pads systems (in both cases the same amount of water was used), showed that pads are a better water-saving solution and they are also better in terms of smaller dry-bulb temperature variation. The advantage of using evaporative pads in comparison to fogging is also the fact that because the air is cooled before its inlet to the building, it may be cleaned from dust [63]. It is also worth mentioning that avoiding higher humidity caused by the operation of this evaporative pad system is currently possible though desiccant segments installation [60].

Some of the aforementioned studies were conducted in pig facilities when only microenvironmental conditions have been taken into consideration [54,62]. However, analysis of the animal body response offers a more complete picture of the effectiveness of the method used. Table 1 presents some literature references regarding both environmental and physiological parameters improved by the application of various cooling technologies in pig farming systems.

Haeussermann et al. [61] and Godyń et al. [53] studied the effect of fogging not only on microclimatic parameters but also on animal production and some physiological markers. Haeussermann et al. [61] found the effect of fogging, performed during the fattening period in hot months, on improvement of average weight gain in growing-finishing pigs. In this study ventilation rate and fogging system operation were automatically controlled by the feedback of the indoor temperature. In turn, Godyń et al. [53] used the simple fogging system running independently of ventilation rate control in one of the two sections of a building for lactating sows. The pipelines with nozzles of this approach were located at the height of the air inlet openings (with no additional fan). The microclimatic parameters, respiratory rate, skin surface and rectal temperature in lactating sows was studied in this research. The main effect of cooling methods was seen in a significant drop of sows’ respiratory rate compared with the females kept in a room without cooling (49.1 vs. 68.6 breaths per minute, respectively).

Direct cooling has also been found to have a positive impact on feed intake [11]. In this study finishing pigs were kept in climatic chambers and the authors tested the effect of three treatments—no cooling, misting just prior to meals and cooling in between meals—on feed intake, meal duration and body core temperature (tympanic). All the pigs were kept at the air temperature of 30 °C. Both misting procedures caused a decrease of body core temperature. The authors claimed that consequences of it led to the higher level of feed intake and longer meal duration. The best effect was observed when animals were subjected to misting just before feeding time.

In the study of Huynh et al. [66], growing-finishing pigs were subjected of two types of simple direct cooling systems—bath and sprinklers. In these systems of maintenance the pigs could go outside or they had no access to outdoor areas. Both bath and sprinkling had a positive effect on respiration rate reduction and skin temperature. Moreover, the pigs kept in pens equipped with sprinkling and without access to the outdoor yard were characterised with the highest weight gain.

The positive effect of directed cooling of pigs’ skin was also found in pigs transported to a slaughterhouse [67,68]. In the first experiment [67], the animals were cooled before transporting and later when unloading. The control group was moved without this kind of treatment. The meat from the experimental group was characterised with lower lactate content and a higher pH value of the longissimus dorsi. In turn, Fox et al. [68] studied the effect of sprinkling on the behaviour and body-core temperature. As in the previous experiment, the skin of the pigs was sprayed with water during loading and unloading. The study also included microclimatic conditions in the semi-trailer and in the slaughterhouse. The body-core temperature was measured using an orally inserted recorder. The device continuously monitored the temperature inside the digestive tract. Cooling the pigs did not statistically affect differences at internal temperature. Nevertheless, when the animals were kept at a temperature exceeding 25 °C, in the sprinkled pigs lower values of body-core temperature were noted. Hence, the authors suggest that the direct cooling could be of great importance in improving thermal comfort in pigs transported at (over 25 °C) higher air temperatures. The test results also showed no difference in animal behaviour when the temperature of the air measured in the semi-trailer exceeded 23 °C. At higher temperatures the water-cooled animals showed a statistically significant tendency to maintain a standing position. The opposite situation occurred in the slaughterhouse where animals had the opportunity to rest. The cooled pigs showed a greater desire to lie down and to drink water less compared with the control individuals.

### 3.2. Technologies Based on High Forced-Air Velocity

Forced convection is one of the most effective ways of improving an animal’s heat loss. In pig facilities the systems may be installed in a way to cool animals in a special zone (e.g., snout cooling) or high air speed over the animal may be achieved in the whole living area (e.g., by tunnel ventilation) [5,69].

Modern buildings may be equipped with a tunnel ventilation system where the operation of a bank of fans mounted at one end of the building causes an increased inflow of air from the other end of the room. Design tunnel barns at about 2 m/s of air speed is a popular solution used in pig maintenance [70]. An optimal cooling effect is achieved by heat transfer through convection, but it may be additionally strengthened by the usage of evaporative cooling methods.

The comparison of the performance parameters in pigs maintained in modern and older types of facilities has been studied [5]. There newer style buildings were: a natural cross-ventilation facility (sidewall curtains), a building with tunnel ventilation and a facility with hybrid ventilation (mechanical ventilation and natural cross–ventilation provided by a side curtain opening). Pigs of other groups were kept in old-style buildings with natural ventilation. In the modern buildings during the hot summer the pigs were additionally sprinkled. The results showed that feed efficiency was significantly better in pigs kept in improved conditions in comparison to older types of facilities.

In the other study [71] the authors evaluated the performance of growing-finishing pigs keeping in buildings with natural ventilation, tunnel ventilation alone and tunnel ventilation combined with evaporative pads (two, cool, cell-pads of a large surface and extractor fans in the opposite wall). All the facilities were curtain-sided buildings. The one of the study results was the finding that keeping animals in both types of mechanically ventilated houses caused higher average daily gain than was shown in individuals maintained in a pig house with natural ventilation (approximately 818 g vs. 793 g, respectively). However there were no differences in the production indicators between pigs cooled by only tunnel ventilation and those cooled by this technology supported with the evaporative cooling method.

A lower level of neutrophil to lymphocyte N/L ratio in peripheral blood (considered as a better marker of exposure to stress in pigs compared to cortisol and cortisone levels in urine) was found in pregnant sows kept in tunnelled, ventilated rooms supported by large evaporative pads. In turn, the higher level of this marker was noticed in sows maintained in a facility when only sprinklers were provided [57].

Zone cooling is another method enhancing animal convective and evaporative heat loss. The cooling of the air stream is especially applicable in individually housed group of pigs. In the study of Dong et al. [72], the operating of a tunnel ventilation system was more effective when this technology was supported both with drip-cooling method alone or drip cooling with vertical head–zone ventilation. The effect of this experiment was seen in a significant reduction of respiratory rate and body-core temperature. It is worth adding that the heat-zone ventilation system in this study was created by using a perforated plastic distribution track connected to fans. The operating of which caused an air velocity of 0.6–0.8 m/s above a sow’s head.

Higher air velocity was used in the other studies [9,69]. In the first-mentioned experiment, the authors found no significant difference between lactating sows cooled with using high air velocity and other methods in values of rectal, skin temperatures and fat thickness. The technologies used in this study were a drip-cooling system alone (drip nozzle supplied 2 L/h) or with a combination of snout cooling (air stream of 7.2 m/s) or drip cooling with the provision of a full-steel sheet placed under the animal’s head. In turn, taking into consideration the females’ behaviour, the authors suggested that the best solution for maintaining the pigs’ comfort (feeling of pleasure) was using the drip-cooling method together with air flow provided by snout-cooling ducts.

In this experiment the sows were kept individually; in turn Barbari and Conti [69] observed the changes in behaviour of females kept in groups. In the described study the pen area was divided into four zones and three of them were equipped with different cooling methods. In the first zone a snout-cooling system providing an air stream of 12.5 m/s flowing towards the floor was installed. In the second zone there were drop-cooling nozzles together with snout-cooling ducts. In the third zone only a drop-cooling system was installed. The authors used the preference test, and behavioural observation showed that at air temperatures of values below 22 °C, sows more often chose a place cooled only by the streamed air and the zone where no cooling system was installed. The situation changed when the air temperatures exceeded 30 °C. The authors found that at higher air temperatures sows preferred the zone in which a combination of cooling by water and the air stream was applied.

An evaporative-cooling pads system is often used to decrease the air-stream temperature in zone-cooling technology [6,10,64,65]. Justino et al. [6] used air of velocity above 3 m/s, cooled by an evaporative-pad system, distributed around group of lactating sows (through plastic ducts with holes in it). This indirect cooling method had a significant impact on respiratory rate reduction (62.5 vs. 46.6 breaths per minute), a decrease in surface-skin temperature (34.7 °C vs. 34.3 °C) and thus it also significantly improved the sows’ sensible heat loss.

In the other experiment [65], comparison of physiological parameters of sows at first, second and third parturition was carried out. The females were kept in a building with natural ventilation alone (cross-ventilated, open side-wall building) or additionally exposed to a cooling system (front part of the animal body subjected to air stream of 10 m/s, additionally supported by evaporative pads). The cooling technology affected an increase of daily ration intake in gilts (36.1 vs. 28.2 g) and in all the tested groups a lower skin-surface temperature (e.g., in gilts 32.9 vs. 34.8 °C) and lower respiratory rate (e.g., in gilts 54 vs. 80 breaths per minute) were found.

Zone-cooling technology associated with evaporative pads was tested also in terms of improving the performance indicators of sows [10,64]. For example, the effectiveness of the evaporative snout-cooling method with a traditional temperature control system (curtain management) on, among others, piglet body weight has been compared [10]. Both the birth and weaning weight of the piglets was higher when sows were kept with cooling technology. In the study of Romanini et al. [64], sows during pregnancy were kept in naturally ventilated rooms (openings in side walls) or in rooms equipped with mechanical ventilation associated with a fogging system. In turn, during the lactation phase, the females were maintained in three types of rooms—naturally ventilated, mechanically ventilated or zone ventilated—with an associated evaporative pad system. In this solution cooled air was pumped to the area above the sows’ head through a PVC tube. Taking into consideration the room for lactating sows, it should be mention that the evaporative-cooled air stream had a significant impact on reduction of respiratory rate in females compared to sows kept in the naturally ventilated room or in the room with mechanical ventilation (50 vs. 54 and 57 breaths per minute, respectively). However taking into consideration piglet weight at weaning, the authors came to the conclusion that the best strategy for maintaining sows in conditions of good comfort and productivity is keeping them in a naturally ventilated room during pregnancy, and then in rooms with zone cooling supported by an evaporative-cooling pad system.

When discussing the methods of cooling pigs through a decrease of air temperature inside the building, it is worth mentioning that currently energy-saving systems of air treatment based on earth-air heat exchange have been developed [4]. Another solution was proposed by Shah et al. [73]. The authors cooled the air before its ingress to the building by using pipes with circulating water buried in the ground. This earth-to-water heat exchanger had the potential to mitigate heat stress in finishing pigs at the same level as fans and a sprinklers system, but it was characterised with significantly lower electricity and water consumption. It can be concluded that the search for this type of equipment (heat exchangers) and the possibility of better indoor air circulation can significantly contribute to increasing of animal comfort and, on the other hand, to producer cost savings.

### 3.3. Floor Cooling

Taking into consideration the large amount of time pigs spend lying, especially individually kept sows, contact with a cool surface may lead to improved comfort of an animal during hot periods. The developing of a floor cooling system using underground water was used both in open-type and closed pig facilities [74,75]. In the study of Shi et al. [73], the pigs were kept in an open house with a floor cooling system (pump-powered water pipelines under a thick layer of concrete) or without this technology. The authors analysed changes in the behaviour of individuals of both groups. Independently of the rise in ambient temperature (23 to 34 °C), 60% of the pigs with access to the cooling area spent their time lying. This was the opposite to the animals of the other group; when the air temperature increased to 30 °C, only 57% of the pigs were lying comfortably and at 30–33 °C, only 10–20% of pigs were found to lie in the sleeping area. No pigs were observed in this area when the ambient temperature increased up to 33 °C; instead the animals were lying or standing in the play area where access to water was provided.

Based on a preferential test, it was found that gilts most often chose the conductive cool-pads to rid themselves of heat load compared to the drip-cooling or snout-cooling methods [76]. In the other study not only behaviour but also production performance and physiological parameters were evaluated [77]. In this experiment one group of lactating sows was kept in individual pens equipped with plates with circulating water (17 °C). In comparison to the females maintained in the typical pens, the cooled sows were characterised with higher feed intake (6.47 vs. 5.60 kg/day, respectively), milk production (10.20 vs. 8.05 kg/sow/day, respectively) and consequently higher weight of piglets at weaning (6.42 vs. 5.30 kg, respectively). With regard to behaviour, it may be stated that the cooled females spent less time sleeping and more time nursing. Moreover, they were eating and drinking more and there was less frequency to urinate compared to the individuals kept uncooled. Taking into consideration indicators such as respiratory rate, skin surface and rectal temperature, the values were generally significantly lower in sows kept with the cooled floor system.

In the above-mentioned study [77], air temperatures were within the range of 20.8 °C to 26.9 °C. However, the other scientists tested the effectiveness of conductive cooling in sows when the ambient temperature increased to a value of 35 °C [8,75,78]. It was found that conductive-cooling pads (water at 18 °C) was pumped through pads at a flow rate of 4 L/min affected lower respiratory rate during acute heat stress exposition [78]. Mean breaths per minute was still at the high level in the cooled sows (83 breaths per minute), however, in comparison to individuals kept in the pens with pads without water circulation, prevented the onset of panting (mean breaths per minute in this group was 107). Moreover the cooling method reduced heat and moisture production by about 10% and 35%, respectively.

In the studies of Cabezón et al. [8] and Parois et al. [75], different flow rates pumped through the pads were tested in lactating sows’ exposure to acute heat stress. Both the experiments had similar methodological assumptions. The sows were located in pens with conductive cooling pads. The first group was kept in the pens with pads without water circulation; the second group was provided with pads of 0.25 L/min water flow; the third group of sows had access to pads of 0.55 L/min water flow and the last group was cooled with water circulation of 0.85 L/min. In the study of Cabezón et al. [8], all the animals were subjected to heat stress conditions of 35 °C air temperature for 90 min. Respiration rates, rectal temperature and skin temperature (15 cm posterior to the ear) were measured. The results showed that during acute heat exposition 0.25 and 0.85 L/min water flow had a significant impact of reduction of respiratory rate and body-core temperature. When compared, the group kept in pens with pads without water flow and with the highest level flow, at the end of the trial the respiratory rate was 132 and 31 breaths/min, respectively. In turn, values of rectal temperature in both discussed groups was 39.9 °C and 39.1 °C, respectively. In turn, in the study of Parois et al. [75] sows were exposed to acute heat stress for 100 min. The authors evaluated the different water flow through the cooling pads on animals behaviour and heart rate. The effect showed that in comparison to individuals of the group kept in pens with pads without water flow, the sows cooled by water circulations (regardless of water flow rate) were characterised by better comfort observed through maintaining a lateral lying position for a longer time and through a lower heart rate.

The aim of the other study [79] was to evaluate the effect of water flow (0.25 or 0.55 L/min) through cooling pads throughout the entire lactating phase. The control group was maintained without water circulation in the installed pads in the floor. The authors evaluated the influence on the animal body of moderate heat stress (air temperature from 27 °C to 32 °C) or mild (22 °C to 27 °C). This long-lasting experiment also confirmed the purposefulness of using this cooling technology, as the authors found a reduction of respiratory rate, body core and skin temperature. As the scientists claimed, when the air temperature rose above 27 °C, water flow of the pad at the level >0.25 L/min is required.

It may be generally concluded that in terms of modern sows at the farrowing and lactating stage, there are opinions that direct cooling with the use of conductive pads is a better solution than lowering the air temperature in the whole building. This way of providing individual cooling for females of different litter sizes and different lactation stages seems to have more advantages for animal welfare and production [74].

It is also worth adding that not only the application of technologically advanced conductive cooling technologies may bring about benefits. As it was showed in finishing pigs that cooling of a partly solid floor was effective in reducing respiratory rate and increasing the time the animals spent lying in a comfortable position [80].

## 4. Conclusions

An increase in the frequency of hot periods which has been observed over the past decades is a growing challenge in providing livestock with adequate living conditions. The choosing of a proper cooling technology depends mainly on the type of pig-farming system as well as the model of facilities. This review has shown that even the older types of building may be equipped with a cooling technology such as sprinkling, fogging system, zone cooling or conductive-cooling pads. The decrease of air temperature before its entry into the building, via the use of smaller surface evaporative pads, is also a universal solution. It seems that the evaporative-cooling pads system combined with high forced-air movement is one of the most effective methods positively affecting the indicators of pig welfare, regardless of the technological group. Moreover, as the results have indicated, this technology may lead to an improvement of psychological and performance parameters in pigs even in countries with climate of high air humidity (Table 1). Respiratory rate remains the most commonly used marker in the research of animal comfort during heat load challenge. However, taking into consideration the current development of diagnostic techniques based on the achievements of molecular biology, there is a justified demand for further research using the novel markers of animal-body response in evaluating cooling strategies. Moreover, the results of recent studies have also illustrated the growing interest in elaborating the new energy- and water-saving cooling solutions which would be applicable in livestock facilities. The development of technologically advanced devices of air treatment and its distribution is definitely a field of future studies on improving animal comfort.

## Figures and Tables

**Table 1 animals-10-01459-t001:** Impact of cooling methods on climatic and physiological parameters in pigs.

Animal/Type of Facility	Cooling Technology	Climatic Parameters		Main Effect on Physiological/Performance Markers	Literature Reference
		Uncooled Room	Cooled Room		
Lactating sows/concrete building with open air inlets	Fogging system	AT 27.8 °C RH 49.8% THI 75.3	AT 25.7 °C RH 68.4% THI 74.5	Decrease of respiratory rate	Godyń et al. [53]
Lactating sows/open side walls building	Evaporative pads with forced ventilation	AT 26.2 °C RH 69.8% AV 0.1 m/s	AT 24.1 °C RH 79.5% AV 3.25 m/s	Decrease of respiratory rate and skin surface temperature	Justino et al. [6]
Lactating sows/open side walls building	Evaporative zone cooling	AT 25.0 °C RH 75.0%	AT 24.3 °C RH 92.5%	Decrease of respiratory rate and increase of back fat thickness	Romanini et al. [64]
Lactating sows/open side walls building	Evaporative zone cooling	AT 29.0 °C RH 60.4% THI 77.4	AT 29.1 °C RH 58.6% THI 76.6	Increase of feed intake (gilts), decrease of respiratory rate and skin surface temperature	Watanabe et al. [65]

AT—air temperature, RH—relative humidity, THI—thermal humidity index, AV—air velocity.

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
