# Peer review of "Use of Different Cooling Methods in Pig Facilities to Alleviate the Effects of Heat Stress—A Review"

_animals, 2020, doi:10.3390/ani10091459_

Round 1

Reviewer 1 Report

The manuscript entitled “Use of different cooling methods in pig facilities to alleviate the effects of heat stress - a review” aimed to present the actual state of knowledge of the most widely used methods of cooling pigs and to show the effectiveness of these technologies on improvement of animal welfare indicators. The article is rich in content and comprehensive in description. Thermoregulation, behavioural and physiological changes in pigs affected by high ambient temperature, and cooling technologies are well summarized, and according to the current research situation, the shortcomings and the future research direction are pointed out, which is of great guiding significance to the industry. I think there are only a few minor problems that need to be revised here, and then acceptable.

  1. Heading 2“Thermoregulation, behavioural and physiological changes in pigs affected by high ambient temperature” should be subdivided to make it look more organized and less cluttered.
  2. Line 458: “ Conclusion”should be “4. Conclusion”.

Author Response

Reviewer 1.

            Firstly we would like to thank the Reviewer for such positive opinions. We hope that our corrections fully correspond to the Reviewer expectations.

            We have divided the chapter about thermoregulation – according to the Reviewer comment. Thank you for this remark as now the text  seems  clearer and more readable. We corrected the number -conclusion. All changes linked to the Reviewer opinions have been marked with yellow color.

Reviewer 2 Report

Dear authors,

I would congratulate on you for this valuable piece of work which I think was needed both for scientists and professionals of the field.

I reckon that a big effort was done to build this review.

Meanwhile I would suggest authors to improve the display of information throughout the manuscript, which is to say to change the scheme.

For instance, a review is much more readable when instead of reporting 'Smith and Jones report that pigs in farms with cooling systems are happier in certain period of the year...' would be easier if saying 'pigs are happier if farms possess cooling systems in certain periods of the year (Smith and Jones)'... just an example. There some tracts in the manuscript that appears as a a series of reporting what other authors say (which is correct, don't misunderstand me...it is perfect), but you may change the style, by having the benefit of a flow of concepts instead of other scientists' list of concepts.

One last thing. Please, stress (sorry for this word trick) on the importance that the improvement of pig health is the first issue and production performance are secondary to animal health. Please, also, or you use pig farming system or pork production system. I would be tempted to use pig farming system throughout the manuscript but not mixing into pig production. They are different phases.

Thank you. Good job.

Author Response

Reviewer 2.

            We would like to thank the Reviewer  for the opinions that are so favorable to our work.

We have made changes which, we hope, correspond to the Reviewer's remarks. We decided to keep a logical sequence of descriptions, (we cared from the beginning to keep the connections with different authors findings in some logical line) however, we significantly reduced the references to specific names of authors.

            The suggestion that welfare and animal health is priority was placed in the first words of the introduction. We decided to add in some places- pig farming system.

            All changes corresponding with the Reviewer comments have been marked with blue. 
            Thank you once again for the words and opinions which will definitely motivate us to the further actions in the field of science.